# Current Applications and the Future of Phage Therapy for Periprosthetic Joint Infections

**DOI:** 10.3390/antibiotics14060581

**Published:** 2025-06-06

**Authors:** Arian Ocean Abedi, Armita Armina Abedi, Tristan Ferry, Mustafa Citak

**Affiliations:** 1Department of Orthopaedic Surgery, Helios Endo-Klinik Hamburg, 22767 Hamburg, Germany; 2Department of Clinical Medicine, Faculty of Health and Medical Sciences, University of Copenhagen, 2200 Copenhagen, Denmark; 3Service de Maladies Infectieuses et Tropicales, Hôpital de la Croix-Rousse, Hospices Civils de Lyon, 69002 Lyon, France; 4Faculty of Medicine, Université Claude Bernard Lyon 1, 69100 Villeurbanne, France; 5Education and Clinical Officer of the ESCMID Study Group for Non-Traditional Antibacterial Therapy (ESGNTA), 4051 Basel, Switzerland

**Keywords:** bacteriophage, phage therapy, periprosthetic joint infection, antimicrobial resistance

## Abstract

Periprosthetic joint infections (PJI) present significant challenges in orthopedic surgery, largely due to the complexity of treating antibiotic-resistant infections. Phage therapy, which utilizes bacteriophages to target bacterial pathogens, offers a promising supplement to traditional antimicrobial methods. This review discusses the current applications of phage therapy in the management of PJI, exploring its underlying mechanisms, clinical outcomes, and practical considerations. We also explore advances in phage therapy technology, including the development of phage cocktails, bioengineered phages, and combination therapies with antibiotics, which enhance the specificity and effectiveness of treatments. Furthermore, we address the future potential of phage therapy to be integrated into standard treatment protocols, focusing on ongoing innovations and research areas.The regulatory and ethical aspects of phage therapy in clinical settings are also discussed. By offering a comprehensive evaluation of both the current state and prospects of phage therapy, this review aims to inform clinical practice and stimulate further research into this innovative treatment modality for PJI management.

## 1. Introduction

Periprosthetic joint infection (PJI) is one of the most challenging complications following total joint arthroplasty (TJA) and is associated with increased morbidity, mortality, and socioeconomic costs [1,2,3,4,5]. Not only is it devastating for patients, but PJI poses significant challenges for orthopedic surgeons due to its persistence with regard to recurrence rates and resistance to antimicrobial treatments [6,7]. This challenge is particularly prevalent in chronic infections, where bacterial biofilms form over prosthetic materials. Biofilms are structured communities of bacteria encased in a self-produced extracellular polymeric substance matrix that adhere to surfaces, including implanted medical devices and prosthetic joints [8]. These biofilm-associated infections are notoriously difficult to eradicate due to the protective environment they offer to bacterial cells [8,9,10]. As a result, traditional treatments often involve a combination of prolonged antibiotic therapy and surgical interventions, including implant removal and staged revision procedures [11]. These strategies, however, are associated with high healthcare costs and suboptimal patient outcomes [12,13]. Therapy with bacteriophages has emerged as a promising supplement in the treatment of antimicrobial-resistant and biofilm-associated infections [14]. Phages are naturally occurring viruses that selectively infect and lyse bacterial cells, providing a targeted therapeutic approach [15]. Unlike antibiotics, phages may co-evolve with bacterial hosts [16]. This offers a potential advantage in circumventing bacterial resistance mechanisms. Recent advances in phage research have demonstrated their efficacy in disrupting biofilms, targeting intracellular pathogens, and synergizing with antibiotics, making them a compelling option for treating PJIs [17,18].

The growing global health crisis of antibiotic resistance has reignited interest in phage therapy as a potential alternative or complementary measure to traditional antibiotics [19,20]. Multidrug-resistant bacteria, including *methicillin-resistant Staphylococcus aureus* (MRSA), a prevalent pathogen in PJIs, pose significant challenges to healthcare systems worldwide [21,22]. Given the limitations of conventional treatment strategies, phage therapy has emerged as a promising approach due to its targeted bacterial eradication. In this review, we aim to evaluate the current state and prospects of phage therapy for PJI management, highlighting its underlying mechanisms, clinical applications, technological advancements, and challenges.

## 2. Phage Mechanisms

Bacteriophages, or phages, are viruses that specifically infect and lyse bacterial cells, playing a crucial role in regulating bacterial populations; these are called lytic phages [23]. However, not all phages are lytic; many follow a lysogenic cycle, integrating their genome into the bacterial host and remaining dormant until triggered [24]. This integration potential demands careful consideration, as lysogenic phages may contribute to horizontal gene transfer or unintended genetic modifications, which may potentially lead to bacterial resistance or other adverse effects. The ability of phages to selectively target bacterial pathogens has made them a promising tool, particularly in the fight against antibiotic-resistant bacteria. Structurally, most phages exhibit an icosahedral head, which houses their genetic material, and a tail structure, which is often contractile and facilitates bacterial attachment and genome injection [25]. However, there is substantial morphological diversity, including non-contractile tails, filamentous forms, and tailless variants [26,27]. The tail fibers or receptor-binding proteins at the end of the tail provide phage specificity, ensuring infection of only particular bacterial species or strains [28]. Once attached, phages inject their genetic material into the bacterial cell, hindering bacterial replication. Depending on the type of phage, this process follows two main pathways: the lytic cycle, where the phage rapidly replicates and lyses the bacterial cell to release new viral particles, or the lysogenic cycle, where the phage genome integrates into the bacterial cell and remains dormant until triggered into lytic activity [29].

Beyond direct bacterial killing, phages demonstrate several key advantages over traditional antibiotics. They can effectively penetrate and disrupt biofilms, which are structured bacterial communities encased in a protective extracellular matrix that makes them highly resistant to antibiotics [30,31]. Phage-derived enzymes degrade this matrix, weakening biofilms and increasing bacterial susceptibility to both phage attack and antibiotics [32,33]. Additionally, phages have shown potential in targeting intracellular bacterial pathogens, which evade traditional antibiotics by residing within host cells [34]. Advances in phage engineering are exploring ways to modify phages to enter host cells and selectively eliminate intracellular bacteria [18]. Another crucial advantage is phage-antibiotic synergy, where certain phage-antibiotic combinations enhance bacterial clearance beyond what either treatment could achieve alone [35,36,37].

Phages also exhibit a self-replicating and self-limiting nature, as they only proliferate in the presence of their bacterial host, making them an adaptive therapeutic tool capable of evolving alongside bacteria [16,38]. Given these unique capabilities, phage therapy represents a promising strategy for addressing PJI and other biofilm-associated bacterial diseases, and in particular those caused by multidrug-resistant pathogens.

## 3. Current Applications of Phage Therapy

### 3.1. Prospective and Comparative Studies

Although this is a narrative review, we conducted a targeted literature search in PubMed and Embase using the terms *phage* OR *bacteriophage* AND *prosthetic joint infection* OR *periprosthetic joint infection*. The search included articles published up to 1 February 2025. Studies were eligible for inclusion if they reported on the use of bacteriophages or related biological agents (e.g., lysins) in the treatment of prosthetic joint infections, including both primary clinical data and systematic reviews. We included studies regardless of design or route of phage administration to provide a comprehensive overview of the clinical landscape. Reference lists of key articles were also screened to identify additional relevant publications.

The most structured evidence to date remains limited. Fedorov et al. conducted a prospective, non-randomized study comparing 23 patients with prosthetic joint infections (PJIs) treated with adjunctive phage therapy to a historical control group (n = 22) receiving only antibiotics [35]. The primary outcome was infection recurrence, with a follow-up extending to one year. At one-year follow-up, the PJI relapse rate was eight times higher in the control group, thus suggesting a potential benefit of phage therapy in reducing recurrence. Phage therapy was well tolerated, with only mild, transient side effects like fever. Despite these promising findings from a comparative study, several limitations exist, including the use of historical controls, variability in bacterial strains between groups, and the short follow-up period, which may limit the generalizability of its conclusions, warranting further investigations. Furthermore, the study implemented a mix of phages to bone cement without demonstrating whether they survived and without providing pharmacokinetic release of the phages from the bone cement. Nonetheless, these results support the feasibility of integrating phage therapy into PJI management, particularly in refractory cases where standard treatments fail. To the best of our knowledge, there is no data available from clinical trials about the evaluation of phage PK/PD and the evaluation of the immune response following phage local and/or intravenous administrations in patients with PJI.

### 3.2. Case Reports and Series: Therapeutic Approaches and Insights

The most substantial portion of the clinical evidence base for bacteriophage therapy in PJI comes from individual case reports and small case series. Table 1 provides an overview of published clinical cases of phage therapy for prosthetic joint infections, including the recent UK series [39]. While similar tables appear in prior reviews [40,41], this version prioritizes clinically relevant variables such as delivery method, surgical strategy, follow-up duration, and outcomes while omitting detailed phage composition to offer a more focused, outcome-oriented synthesis for clinical application. As outlined in Table 1, these studies provide valuable insights into how phage therapy has been integrated across diverse clinical contexts despite heterogeneity in protocols, pathogens, and phage formulations.

The 2024 systematic review by Yang et al. [40] remains the most comprehensive synthesis to date, including 16 clinical studies with 42 patients treated for PJIs using bacteriophages. In this cohort, *Staphylococcus aureus*, including methicillin-resistant strains, was the most frequently reported pathogen (n = 18), followed by *P. aeruginosa*, *S. epidermidis*, *K. pneumoniae*, *E. faecalis*, and a few polymicrobial infections. Most patients had undergone multiple prior surgeries, with bacteriophage therapy used either as part of a salvage approach or integrated into revision procedures. Phage therapy delivery varied widely. Intraoperative local application was common; this was either through direct flooding of the surgical field or incorporation into carriers such as bone cement. Somecases employed postoperative intra-articular delivery via catheters or drains over 5–14 days. Intravenous administration was used in some cases, often in combination with local delivery. Across almost all reports, phage therapy was paired with systemic antibiotics, typically vancomycin, daptomycin, or ciprofloxacin, reflecting both a synergistic rationale and the real-world need to adhere to antimicrobial standards.

Outcomes across the case studies in the literature were generally positive. Most patients experienced clinical resolution of infection or substantial symptom improvement, often with documented healing of fistulas and absence of local signs of infection at follow-up. Follow-up durations varied but, when reported, ranged from 4 weeks to over one year, with several cases reporting durability of remission beyond 12 months.

The recent UK case series from 2025 [39] represents a valuable addition to this evolving body of literature. This report detailed three patients with PJIs caused by *S. aureus*, *S. epidermidis*, and a polymicrobial infection involving *K. pneumoniae*. Phage cocktails were administered intraoperatively and postoperatively via drain infusion, with two of the three patients achieving complete clinical resolution. Notably, one case involved the development and emergency import of a customized phage for *K. pneumoniae*, illustrating the logistical and regulatory complexities of tailored phage therapy under current frameworks. This case also serves as a reminder of the challenges in polymicrobial or biofilm-associated infections, where bacterial species may respond differently to phage exposure.

Despite promising outcomes, variability in surgical context and treatment protocol limits generalizability. Surgical strategies included DAIR, one- and two-stage revisions, prosthesis explantation with spacer placement, and, in rare cases, amputation. The diversity of surgical interventions makes it difficult to isolate the effect of phage therapy alone. Additionally, most cases involved extensive use of suppressive antibiotic therapy, blurring the line between phage-specific and multimodal treatment outcomes. Importantly, safety outcomes across the reviewed studies were consistently reassuring. Adverse events were rare and generally mild, such as transient fever, chills, or local erythema following initial phage administration. These symptoms were usually self-limiting or resolved with dose adjustment, suggesting an overall favorable tolerability profile.

Taken together, the case-based literature supports the use of phage therapy as a feasible adjunct in the management of PJIs, particularly in complex or refractory cases. While not definitive, these reports provide encouraging evidence that personalized, targeted phage therapy, when administered alongside appropriate surgical and antimicrobial management, can contribute meaningfully to infection resolution, even in settings of multidrug resistance or biofilm persistence.

### 3.3. Synthesis from Reviews

Several reviews have been conducted and converge on promising outcomes but also expose gaps. The aforementioned systematic review conducted by Yang et al. [40] emphasized clinical resolution in 95% of included patients but highlighted a lack of standardization in treatment protocols. In another study, a proportional meta-analysis aimed to provide a preliminary assessment of early phage therapy outcomes for PJI [57]. The study reported an infection remission rate of 78%, with improved durability of outcomes (83% remission) in patients followed for at least 12 months. Most patients had staphylococcal infections, with intraarticular phage delivery being the predominant route (73%). Additionally, phage cocktails (65%) and antibiotic co-administration (97%) were common, suggesting synergy between phage therapy and standard antimicrobial regimens.

Another literature review evaluating infections with *S. aureus* found evidence for phage-antibiotic synergy, especially with vancomycin and daptomycin [41]. A systematic review of the safety and efficacy of phage therapy further confirmed that phages are generally safe, with no severe adverse events reported in modern trials [58]. However, it noted that therapeutic success depends on delivering a sufficient quantity of well-matched phages to the infection site. Studies that failed to demonstrate efficacy often lacked appropriate phage selection, dosing, or delivery strategies. This underscores the importance of individualized, patient-specific phage formulations, particularly in biofilm-associated infections like PJIs.

Furthermore, a recent scoping review analyzed 77 studies on phage therapy in orthopedic infections, reinforcing direct phage delivery as the most commonly applied method and highlighting several knowledge gaps, including optimal dosing, phage pharmacokinetics, and delivery methods, as well as the role of phage cocktails versus monophage therapy [59]. The authors found that the lack of comparative clinical trials was a major limitation in translating phage therapy into widespread clinical practice.

Although study overlaps are significant across these reviews, their thematic emphases differ; some prioritize efficacy metrics, others delivery strategies or biofilm-targeting capacity. Critically, nearly all studies lacked control groups and consistent outcome measures.

### 3.4. Integration into Clinical Practice

Currently, phage therapy is being integrated into structured treatment protocols and clinical trials to standardize its application for PJIs. French initiatives like the PhagoDAIR I clinical study incorporate phage therapy with DAIR to enhance treatment success rates [46]. Similarly, the PHAGEinLYON *Clinic* program personalizes phage therapy through intraoperative administration and post-operative sonography-guided injections, particularly for multidrug-resistant PJIs [60]. Moreover, a French study found that 44% of PJI patients met the criteria for adjunctive phage therapy, reflecting its growing potential in managing complex, antibiotic-resistant infections [61]. The UK case series demonstrates a similarly evolving landscape: phages were used intraoperatively and via postoperative drains, with individualized matching and bespoke production. Phage selection was driven by susceptibility testing and personalized to clinical isolates. It is important to note that adverse events have been reported as mild, with local erythema or transient fever being the most frequently reported symptoms, likely immune responses rather than elicited by the phage particles themselves. While evidence of long-term success is accumulating, large-scale clinical trials are still needed to investigate standardized phage-antibiotic regimens for chronic orthopedic infections, including PJI, with a focus on refining dosing strategies and assessing long-term efficacy. Phage therapy involves multimodal regimens involving antibiotics and surgical intervention, which is why a definitive attribution of treatment success to phage remains challenging despite encouraging signals of efficacy.

Collectively, these case-based and early prospective findings underscore phage therapy’s potential in PJI management, particularly in cases with limited surgical options, antibiotic resistance, or biofilm persistence. However, more robust data are needed. Several clinical trials are currently underway, and their findings will be critical in defining the role of phages in future PJI treatment algorithms.

## 4. Advances in Phage Therapy

Recent advancements in phage therapy have focused on engineering phages, optimizing delivery mechanisms, and enhancing antibiotic synergy. To overcome bacterial resistance, modified phages with enhanced infectivity and broader host ranges are being developed [62]. Engineered phages expressing depolymerases or endolysins have shown improved biofilm degradation, making infections more susceptible to treatment [32,33,63]. Additionally, phage-antibiotic synergy has gained attention, with studies demonstrating enhanced bacterial clearance, particularly in MRSA infections, with studies suggesting that phage pre-treatment can sensitize bacteria to antibiotics, increasing their efficacy [40,41,57].

To improve phage delivery, novel formulations such as phage-loaded hydrogels, nanoparticles, and implant coatings are being explored [64,65,66,67,68]. In PJIs, intra-articular injection, intravenous administration, and phage-infused biomaterials are under investigation to prolong activity at infection sites and improve retention.

## 5. Future Directions and Challenges

Despite promising developments, several challenges must be addressed before phage therapy can be widely implemented for PJIs. Standardized treatment protocols for phage dosing, administration routes, and quality control are currently lacking, complicating regulatory approval. Phase I/II clinical trials exploring the PK/PD of the administered phage or phage cocktail and the potential immune response depending on (i) the type of phage used, (ii) its way of administration, and (iii) its level of exposure is mandatory before performing large-scale clinical trials to establish efficacy and evaluate long-term outcomes. Of note, due to various reasons, including feasibility, previous clinical trials in PJI were withdrawn, including one in the USA (NCT05269134); another one has been completed in France on patients with *S. aureus* PJI with indication of DAIR combined with suppressive antimicrobial therapy (“PhagoDAIR I”), but the results are not available, and another one, a proof-of-concept study to assess the safety and efficacy of phage therapy in *S. aureus* PJI treated by DAIR (GLORIA; NCT06605651), has not yet started.

Bacterial resistance to phages is another concern, as bacteria can develop defense mechanisms similar to antibiotic resistance. Strategies such as phage cocktails, engineered phages, and rotating phage regimens aim to mitigate resistance development. The host immune response presents additional hurdles, as phages may be neutralized before reaching infection sites. Encapsulation techniques and immunomodulatory approaches are being explored to prolong phage activity [69]. Phage stability and storage also present challenges. Phages are sensitive to temperature and pH, necessitating advanced formulation strategies to maintain viability [70,71]. Ensuring stable, scalable production is critical for widespread clinical use.

## 6. Regulatory, Ethical, and Economic Aspects

Phage therapy faces unique regulatory challenges due to its specificity, adaptability, and potential for horizontal gene transfer. Standardized protocols for phage characterization, production, and clinical evaluation are essential for regulatory acceptance. Agencies like the European Medicines Agency (EMA) have developed preliminary guidelines for veterinary applications, but human-use regulations remain in the early stages [72]. Emergency-use pipelines and phage banks, such as the one in Belgium, provide models for controlled access while regulatory frameworks evolve [73]. Notably, Belgium has implemented a unique regulatory pathway based on magistral, also called compounded preparation, which permits pharmacists to compound patient-specific phage treatments under a physician’s prescription [74,75]. These prescriptions must comply with standards from the European and Belgian Pharmacopoeias, with oversight from the Ministry of Public Health and certified laboratories [76]. In cases where the pharmacopoeial guidance is lacking, supplementary directives may be issued by the Minister of Public Health, with quality control overseen by certified Belgian laboratories. This framework, operational since 2016, enables legally sanctioned, quality-controlled phage therapy outside of clinical trials [75]. In the United States of America, the FDA regulates phage therapies as biologics but permits case-by-case access through its Emergency Investigational New Drug (eIND) program [77,78]. Together, these approaches exemplify how adaptive national models are addressing clinical needs during the ongoing development of global phage therapy regulations.

The challenge for phage banks is to have significant numbers of active phages that could be tested on the patient’s strains. There are also very few phages easily available for particular pathogens (coagulase-negative *staphylococci*, *Enterobacter* spp., *Actinomyces* spp., etc.). The other practical issue when treating a patient with phage therapy is also to perform it easily before administration of the phage susceptibility testing. Most of the time, this has been carried out by the phage producer, requiring significant time and financial resources for shipping the bacterial strain and conducting the required experiments.

From an economic standpoint, phage therapy could reduce healthcare costs by preventing implant removal and reducing prolonged antibiotic use. While personalized phage production remains costly, commercial-scale manufacturing and standardized phage libraries may enhance cost-effectiveness for patients infected with the most common pathogens.

## 7. Discussion

The treatment of PJI remains a major challenge in orthopedic surgery and infectious disease medicine, particularly in recurrent cases and in the context of increasing antimicrobial resistance and biofilm-associated persistence. Conventional treatment strategies comprising surgical interventions such as debridement, implant retention, or revision procedures, in combination with prolonged antimicrobial therapy, can be unsuccessful, especially in chronic or recurrent infections. Bacteriophage therapy has re-emerged in recent years as a promising adjunctive treatment modality. Its potential for high specificity, biofilm penetration, and favorable safety profile offers an alternative mechanism of action that could complement or, perhaps in certain cases, surpass the limitations of antibiotics alone.

In this review, we examined the current landscape of phage therapy in the context of PJIs. While there is a growing number of clinical case reports and series highlighting successful outcomes in individual patients with PJI, the overall clinical evidence base remains limited in scale. Nevertheless, the available data suggest that phage therapy, when appropriately matched to the causative organism and applied in a tailored manner, can play a meaningful role in the management of refractory PJIs.

Much of the current evidence supporting phage therapy is derived from particularly individual case reports and small case series where phage preparations have been administered as a salvage therapy following the repeated failure of conventional approaches. These studies often involve patients with persistent or relapsing infections, many of whom are colonized or infected with multidrug-resistant organisms. In this context, the resolution of infection temporally associated with phage therapy, particularly in cases where antibiotics alone had failed, offers a compelling clinical signal. Importantly, phage therapy is frequently accompanied by other interventions, such as surgical debridement or concurrent antibiotic therapy. This complicates the ability to conclude phage efficacy in isolation. However, several of the included studies report in vitro phage susceptibility testing that confirms the targeted activity of the administered phages against the patient’s infecting strain, lending plausibility to the observed outcomes. These points need to be weighed against the fact that only case reports with a positive outcome are generally published and that lessons learned from the failure of phage therapy are neither discussed nor available.

Despite encouraging results, it is essential to recognize that observational studies are subject to a lack of control groups and demonstrate significant heterogeneity in phage preparation, dosing regimens, and outcome measures. Moreover, as all drugs are designed to demonstrate an added value in modern medicine, randomized placebo-controlled trials are mandatory. Nevertheless, the consistency of findings across multiple reports and international settings, particularly in cases of antibiotic failure, suggests that the observed benefits warrant continued structured investigation.

Based on the available literature, one of the most notable features of phage therapy is its favorable safety profile. Across published studies, adverse effects appear to be infrequent and generally mild. These symptoms are often attributed to immune activation rather than to direct toxicity of the phage itself. Phages are biologically dynamic agents. Their specificity, while advantageous from a safety standpoint, poses major logistical challenges for standardization and commercialization. Phage preparations need to have been extensively characterized beforehand (sequencing revealing the absence of insertion sequences, as lysogenic phages capable of integrating their genome into that of bacteria should be avoided, and the absence of resistance or virulence genes) and have to be of pharmaceutical quality, like any treatment administered to humans. Then, phage preparations often require customization to the patient’s specific pathogen(s), especially in the context of polymicrobial infections or rapidly evolving resistance profiles. This necessitates real-time microbiological support, phage susceptibility testing, and sometimes a reformulation of the phage cocktail. This may be difficult to integrate into rigid clinical trial frameworks.

Compounding these challenges is the heterogeneity in clinical phage use across published studies. Differences in phage source, formulation (single versus cocktail), administration route (intra-articular, intravenous, local, flooding), frequency, duration, and concurrent antibiotic therapy make it difficult to draw broad generalizations. Additionally, the current absence of standardized outcome definitions in terms of what constitutes clinical success or recurrence further adds variability to the available data. Moreover, the confounding effect of concurrent antibiotic therapy remains a central issue in interpreting the efficacy of phage therapy. Many patients treated with phages also receive prolonged antimicrobial regimens and often undergo surgical interventions. The improvement of the patient’s clinical conditions after several prior interventions may be linked to phage administration; however, isolating its independent effect remains challenging.

Moving forward, large-scale trials, while ideal in theory, may not be feasible or informative given the personalized nature of phage therapy. Instead, prospective registries, pragmatic clinical trials, and international collaborative networks may offer appropriate frameworks to assess phage therapy’s role in complex infections. Such efforts must, however, include standardized reporting of phage characteristics, administration protocols, and outcome metrics to allow meaningful data synthesis and guideline development.

## 8. Conclusions

Phage therapy presents a promising adjunct for the management of PJIs, particularly in cases involving biofilm formation and antibiotic resistance or surgical ineligibility. While early evidence suggests favorable outcomes, further research and international efforts are needed to refine phage formulations, optimize treatment protocols, and address regulatory and clinical barriers. The literature emphasizes the need for standardized treatment protocols and larger-scale clinical trials. With continued advancements, phage therapy has the potential to become an integral component of PJI infection management, offering a targeted and adaptive approach to combating bacterial persistence in PJI.

## Figures and Tables

**Table 1 antibiotics-14-00581-t001:** Clinical studies with the use of phages for the treatment of PJI.

Study	Country	N	Joint(s)	Pathogens	Phage Matching Performed?	Phage Treatment	Surgery	Outcome
Ferry et al. [42]	France	1	Hip	Multidrug-resistant *Pseudomonas aeruginosa* and Methicillin-susceptible *Staphylococcus aureus*	Yes	Cocktail. Local injection. Systemic antibiotic treatment.	Planned DAIR, mobile component exchange not possible. Number of surgery: six.	Following a repeat DAIR procedure and the addition of ciprofloxacin, the patient demonstrated no clinical signs of persistent infection and no complications at the 18-month follow-up.
Patey et al. [43]	France		Knee	*Pseudomonas aeruginosa*	NR	Cocktail. Local injection.	Type of surgery: NR. Number of surgery: NR.	Clearance of *Pseudomonas aeruginosa* but the appearance of *Enterococcus* sp.
			Hip	Methicillin-resistant *Staphylococcus aureus*, MRSA	NR	Single (local application and via a catheter in the 10 days following the operation)	Type of surgery: NR. Number of surgery: NR.	No recurrence after 1 year with retention of prosthesis and osteosynthesis material in situ.
			Knee	*Staphylococcus* spp.	NR	Single local application	Type of surgery: NR. Number of surgery: NR.	Fistula closure and partial disinfection, followed by stabilization.
Tkhilaaishvili et al. [44]	Germany	1	Knee	Multidrug resistant *Pseudomonas aeruginosa*	Yes, phage matching performed using isolate-specific selection from phage collection	Single intraoperative loading dose, followed by 5 mL phage every 8 h via four drains for 5 days, plus 6 weeks of systemic antibiotics	Explantation and debridement. Number of surgery: four.	Favorable recovery after debridement and spacer exchange 2 weeks post-exploration and prosthesis reimplantation after 4 weeks. No complications. Follow-up was not reported
Doub et al. [45]	USA	1	Knee	Methicillin-resistant *Staphylococcus aureus*, MRSA	Yes	Single (two intraarticular doses + three IV doses starting the next day). Systemic antibiotic treatment for 6 weeks	Prosthesis explant with placement of static spacer containing vancomycin and tobramycin. Number of surgery: six.	Intraoperative cultures negative; discharged after 1 week. Follow-up was not reported.
Ferry et al. [46]	France	3	Knee	Methicillin-susceptible *Staphylococcus aureus*, MSSA	Yes	Cocktail (PhagoDAIR procedure *) plus systemic antibiotic treatment for 3 months.	DAIR. Number of surgery: two.	New DAIR performed at 3 months; outcome favorable at 30-month follow-up. No complications.
			Knee	Methicillin-susceptible *Staphylococcus aureus*, MSSA	Yes	Cocktail (PhagoDAIR * procedure) plus systemic antibiotic treatment for 3 weeks	DAIR. Number of Surgery: five.	No signs of infection at 7-month follow-up. No complications.
			Knee	Methicillin-susceptible *Staphylococcus aureus*, MSSA	Yes	Cocktail (PhagoDAIR procedure *) plus systemic antibiotic treatment.	DAIR. Number of surgery: three.	New DAIR performed at 4 months. No infection was diagnosed at 11-month follow-up. No complications.
Ferry et al. [47]	France	1	Knee	Methicillin-susceptible *Staphylococcus aureus*, MSSA	Yes	Cocktail. DAC^®^ hydrogel mixed with sterile water and bacteriophage cocktail (1 mL of each phage). Systemic antibiotic treatment.	DAIR + the DIEP free flap. Number of surgery: four.	Two DAIR procedures post-phage; transfemoral amputation at 1 year. Follow-up: 12 months. No complications to phage therapy were reported.
Cano et al. [48]	USA	1	Knee	*Klebsiella pneumoniae complex*	Yes	Single. Daily infusions for a total of 40 doses. Systemic antibiotic infusion.	No surgery. Most recent surgery: Incision and drainage. Number of surgery: 14.	Symptoms resolved; able to perform daily activities to some extent, with a follow-up of 8.5 months. No complications were reported.
Doub et al. [49]	USA	1	Knee	Multidrug resistant *Staphylococcus epidermidis*	Yes	Single. injected into the intraarticular space. Systemic antibiotic treatment for 6 weeks.	DAIR. Number of surgery: five.	No clinical recurrence at 5-month follow-up. Transient liver function abnormality noted on postoperative day 2.
Ferry et al. [50]	France	1	Knee	*Pseudomonas aeruginosa*	Yes	Cocktail. Injected through the arthroscope. Systemic antibiotic treatment > 6 months.	Arthroscopic DAIR. Number of surgery: two.	Resolved pain during motion. Twelve-month follow-up. No complications to phage therapy.
Neuts et al. [51]	The Netherlands	1	Hip	*Enterococcus faecalis*	Yes, isolate specific susceptibility testing on commercial phage cocktails	Cocktail. Oral suspension. Systemic antibiotic therapy.	No surgery. Number of prior surgeries: eight.	At 36-month follow-up, no complaints and no new cultures obtained after treatment. No complications.
Rairez-Sanchez et al. [52]	USA	1	Knee	Methicillin-susceptible *Staphylococcus aureus*, MSSA	Yes	Cocktail. Cycle 1: Single intra-articular dose + IV every 12 h for 2 weeks; Cycle 2: Intraoperative dose + IV every 12 h for 6 weeks. Systemic antibiotic treatment.	Two-stage exchange. Number of surgeries > six.	Negative bacterial culture and stable weekly labs at 14-month follow-up. No complications.
Schoeffel et al. [53]	USA	1	Hip and Knee	Methicillin-resistant *Staphylococcus aureus*, MRSA	Yes	Single. Intra-articular bacteriophage injection followed by daily IV bacteriophage for 3 days postoperatively. Systemic antibiotic therapy.	Single-stage exchange hip and knee. Number of surgery: four.	No recurrence at 11-month follow-up. The patient retained normal function. Mild liver function abnormalities on postoperative day 1 after phage therapy, with no further deterioration.
Racenis et al. [54]	Latvia	1	Hip	Multidrug resistant *Pseudomonas aeruginosa* Vancomycin-resistant *enterococci* and *Staphylococcus epidermidis*	Yes	Cocktail: Intraoperative wound rinsing, followed by local irrigation three times daily for 7 days, then twice daily for another 7 days via a catheter. Systemic antibiotic treatment.	Two-stage exchange. Number of surgery: nine.	No local sign of infections at 15-month follow-up. Acute kidney injury related to prior treatment with meropenem and colistin.
Cesta et al. [55]	Italy	1	Hip	*Pseudomonas aeruginosa*	Yes	Single, 10 mL on day 1, then 5 mL via joint drain for 2 weeks. Twelve weeks of systemic antibiotic treatment.	DAIR and mobile component exchange. Number of surgery: three.	No local signs of infection relapse at 24-month follow-up. Fever and chills after the first dose; resolved with dose reduction.
Fedorov et al. [35]	Russia	23	Hips	*Staphylococcus epidermidis*, MSSE: 8 *Staphylococcus**epidermidis*, MRSE: 6 *Staphylococcus aureus*, MSSA: 8 *Staphylococcus aureus*, MRSA: 1	No, only spot-assay used to assess susceptibility to commercial phage preparation	Cocktail. Phage was added to bone cement, and 20 mL was injected daily via drain for 10 days. Systemic antibiotic treatment, IV for 2 weeks, followed by personal treatment.	One-stage revision. Number of prior surgeries NR.	One case of PJI relapse at 12-month follow-up. Two patients with febrile complications at phage preparation. A combination of phage and antibiotic therapy was found to be more effective than phage alone.
Doub et al. [56]	USA	1	Knee	*Enterococcus faecalis*	Yes	Single. Intra-articular injection via arthrocentesis for 2 days, followed by 4 days of IV phage therapy. Six weeks of systemic antibiotic treatment.	No surgery. Number of prior surgeries > three.	No clinical signs of infection relapse at 24-month follow-up.
Munteanu et al. [39]	UK	3	Hip	Methicillin-sensitive *Staphylococcus aureus* (MSSA)	Yes	Single. Washout with suspension of phage followed by administration through drain three times a day for a total of four days post-surgery. Systemic antibiotic therapy.	Surgical debridement. Number of surgery: four.	Two weeks post-op: repeat washout/debridement. Cultures positive for *P. aeruginosa*, MSSA negative. Antibiotics adjusted. At 9-month follow-up: wound healed, CRP normal, patient walking independently. Non-irritable erythema around the surgical wound after phage therapy resolved. The fever on the fourth day of drain therapy caused the discontinuation of phage therapy; fever resolved.
			Hip	*Klebsiella pneumoniae*, *Corynebacterium striatum*, and methicillin-sensitive *Staphylococcus aureus (MSSA)*	Yes	Cocktail. Intraoperative local application after surgical washout and seven postoperative doses via drain over 4 days. Systemic antibiotic therapy.	Surgical debridement. Number of surgery: five.	Post-op fever and nausea resolved. The wound reopened and persisted, requiring further washout and partial hardware removal at 2 months. Cultures negative; PCR positive for *K.* pneumoniae and MSSA. Completed 12-week antibiotics; wound closed and dry at discharge.
			Knee	*Staphylococcus epidermidis*	Yes	Single. At the time of spacer removal, the joint was washed out with the phage suspension. Systemic antibiotic therapy.	Spacer removal and reimplantation. Number of surgery: three.	At 6-month follow-up. Pain-free and wound completely healed. No complications to phage therapy.

Abbreviations: DAIR, debridement, antibiotics and implant retention; IV, intravenous; PJI, periprosthetic joint infection, DAC^®^: drug-administering composite hydrogel; NR: not reported. * PhagoDAIR procedure: 1 mL of 1 × 1010 PFU/mL for each phage as “magistral” preparation administered by the surgeon directly into the joint after the DAIR procedure and joint closure

## Data Availability

Not applicable.

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
