# Peer review of "Current Applications and the Future of Phage Therapy for Periprosthetic Joint Infections"

_antibiotics, 2025, doi:10.3390/antibiotics14060581_

Round 1
Reviewer 1 Report
Comments and Suggestions for Authors
In this review by Abedi et al., the authors present an overview of the current literature on periprosthetic joint infection (PJI), specifically in the context of phage therapy as a potential treatment. The authors summarize previous research studies and reviews, reiterating well-known challenges such as treatment failure due to biofilm formation and antimicrobial resistance, while highlighting the growing interest in phage therapy as an adjunctive option.
Although the review does not introduce new primary data or conduct any meta-analysis, it draws attention to several important yet often under-discussed aspects of phage therapy, including the significance of pathogen-specific phage matching, the consistent reports of clinical success across varied healthcare settings, and the limitations of applying conventional randomized controlled trial designs to personalized phage therapy.
Overall, while the review is well-written, it falls short in offering new insights into the practical application of phage therapy for PJI. Notably, it lacks discussion on the role of host immunity in shaping phage efficacy and does not explore how current knowledge could guide strategies to overcome immunogenicity and antibody-mediated neutralization. Furthermore, the review would benefit from a deeper analysis of key unresolved issues such as phage pharmacokinetics in joint infections and the integration of phage therapy into existing treatment algorithms.
Author Response
Reviewer 1.
Comment 1: In this review by Abedi et al., the authors present an overview of the current literature on periprosthetic joint infection (PJI), specifically in the context of phage therapy as a potential treatment. The authors summarize previous research studies and reviews, reiterating well-known challenges such as treatment failure due to biofilm formation and antimicrobial resistance, while highlighting the growing interest in phage therapy as an adjunctive option.
Response 1: We would like to thank the reviewer for their thoughtful summary and positive assessment of our manuscript. We appreciate the recognition of our efforts to provide a comprehensive overview of current challenges in PJI treatment and the potential role of phage therapy in addressing this issue.
Comment 2: Although the review does not introduce new primary data or conduct any meta-analysis, it draws attention to several important yet often under-discussed aspects of phage therapy, including the significance of pathogen-specific phage matching, the consistent reports of clinical success across varied healthcare settings, and the limitations of applying conventional randomized controlled trial designs to personalized phage therapy.
Response 2: We thank the reviewer for highlighting these key points. We appreciate the recognition of our aim to draw attention to nuanced and often overlooked aspects of phage therapy.
Comment 3: Overall, while the review is well-written, it falls short in offering new insights into the practical application of phage therapy for PJI. Notably, it lacks discussion on the role of host immunity in shaping phage efficacy and does not explore how current knowledge could guide strategies to overcome immunogenicity and antibody-mediated neutralization. Furthermore, the review would benefit from a deeper analysis of key unresolved issues such as phage pharmacokinetics in joint infections and the integration of phage therapy into existing treatment algorithms.
Response 3: We thank the reviewer for this comment, as it is indeed an important point. We added line 124 to 127 to in the « prospective and comparative studies » section the following sentence: To the best of our knowledge, there is no data available from clinical trials about the evaluation of phage PK/PD and about the evaluation of the immune response following phage local and/or intravenous administrations in patients with PJI. We also added in the « future direction section » line 261 to 265: Phase I/II clinical trials exploring PK/PD of the administered phage of phage cocktail, and the potential immune response depending on: (i) the type of phage used; (ii) its way of administration; and (iii) its level of exposure, are mandatory before performing large-scale clinical trials to establish efficacy and evaluate long-term outcomes.
Reviewer 2 Report
Comments and Suggestions for Authors
The manuscript provides a comprehensive and timely review of bacteriophage therapy for periprosthetic joint infections (PJIs), a topic of growing relevance given the challenges posed by antimicrobial resistance and biofilm-associated chronic infections. The paper is well-structured and scientifically grounded, offering a balanced discussion on current clinical evidence, experimental progress, and challenges for clinical translation.
The flow of the manuscript is logical and intuitive, with clear subheadings and cohesive narrative.
The review is primarily narrative in nature, yet it includes a table with detailed clinical outcomes. Consider including a brief methods section that explains how literature was selected for inclusion (e.g., search strategy, time range, inclusion/exclusion criteria), even if this is not a systematic review.
The paper briefly mentions ongoing clinical trials and registry approaches. It would be beneficial to list any currently registered trials (e.g., ClinicalTrials.gov or EU Clinical Trials Register) to inform the reader of what evidence may be on the horizon.
Table 1: Consider adding columns for “Route of administration” and “Phage matching performed?” to enhance the clinical interpretability of the table.
Author Response
Comment 1: The manuscript provides a comprehensive and timely review of bacteriophage therapy for periprosthetic joint infections (PJIs), a topic of growing relevance given the challenges posed by antimicrobial resistance and biofilm-associated chronic infections. The paper is well-structured and scientifically grounded, offering a balanced discussion on current clinical evidence, experimental progress, and challenges for clinical translation.
Response 1: We sincerely thank the reviewer for their kind and encouraging feedback. We are pleased that the manuscript was found to be well-structured and scientifically grounded. We appreciate the recognition of our efforts to provide a balanced and comprehensive overview of phage therapy in the context of periprosthetic joint infections.
Comment 2: The flow of the manuscript is logical and intuitive, with clear subheadings and cohesive narrative.
Response 2: We appreciate the reviewer’s positive comment on the structure and flow of the manuscript. Thank you.
Comment 3: The review is primarily narrative in nature, yet it includes a table with detailed clinical outcomes. Consider including a brief methods section that explains how literature was selected for inclusion (e.g., search strategy, time range, inclusion/exclusion criteria), even if this is not a systematic review.
Response 3: We thank the reviewer for this comment. In response, we have added a brief section in the beginning of section 3.1. «prospective and comparative studies», to clarify our literature selection process we added the following at lines 101 to 109: Although this is a narrative review, we conducted a targeted literature search in PubMed and Embase using the terms phage OR bacteriophage AND prosthetic joint infection OR periprosthetic joint infection. The search included articles published up to February 1st, 2025. Studies were eligible for inclusion if they reported on the use of bacteriophages or related biological agents (e.g., lysins) in the treatment of prosthetic joint infections, including both primary clinical data and systematic reviews. We included studies regardless of design or route of phage administration to provide a comprehensive overview of the clinical landscape. Reference lists of key articles were also screened to identify additional relevant publications. We hope this addition enhances transparency and helps explain the studies presented in the review.
Comment 4: The paper briefly mentions ongoing clinical trials and registry approaches. It would be beneficial to list any currently registered trials (e.g., ClinicalTrials.gov or EU Clinical Trials Register) to inform the reader of what evidence may be on the horizon.
Response 4: We thank the reviewer for this valuable suggestion and the opportunity to elaborate on ongoing clinical research. We have added the following sentence in the « future direction section » lines 265 to 270: Of note, due to various reasons including feasibility, previous clinical trials in PJI were withdrawn, including one in the USA (NCT05269134); another one has been completed in France patients with S. aureus PJI with indication of DAIR combined with suppres-sive antimicrobial therapy (“PhagoDAIR I”), but the results are not available, and an-other one, a proof-of-concept study to assess safety and efficacy of phage therapy in S. aureus PJI treated by DAIR (GLORIA; NCT06605651), has not yet started.
Comment 5: Table 1: Consider adding columns for “Route of administration” and “Phage matching performed?” to enhance the clinical interpretability of the table.
Response 5: We thank the reviewer for this thoughtful suggestion. The route of administration is already described within the “Phage treatment” column of Table 1, but to improve clarity, we have now added further details on administration for the PhagoDAIR study in a footnote lines 141 to 142: *PhagoDAIR procedure: 1 ml of 1 × 1010 PFU/ml for each phage as “magistral” preparation administered by the surgeon directly into the joint, after the DAIR procedure and joint closure. Additionally, we have added a new column to indicate whether phage matching was performed: "Phage Matching Performed?", as suggested. We hope these additions sufficiently address the reviewer’s suggestion, or that the existing presentation of the route of administration within the “Phage treatment” column will be acceptable.
Reviewer 3 Report
Comments and Suggestions for Authors
The authors describe current knowledge regarding phage therapy for prosthetic joint infections, an attractive strategy for recalcitrant and difficult-to-manage infections.
Lines 61-66: “Bacteriophages, or phages, are viruses that specifically infect and lyse bacterial cells,playing a crucial role in regulating bacterial populations [23]. Their ability to selectively target bacterial pathogens has made them a promising tool, particularly in the fight against antibiotic-resistant bacteria. Structurally, most phages exhibit an icosahedral head, which houses their genetic material, and a tail structure, which is often contractile and facilitates bacterial attachment and genome injection”
Comment: While describing the phage mechanisms, the authors should detail that not all existing phages are lytic; many are lysogenic. And although the typical structure is as described, even the lytic ones may present various structures.
Table 1:
Comment: DAIR and other acronyms should be defined in the table legend. If possible, the source and name of the phages used should be added if available from the original paper.
If available, cases of unsuccessful phage treatment should be included.
Staphylococcus Aureus, should be corrected as Staphylococcus aureus
Staphylococcus sp, should be corrected as Staphylococcus spp.
Lines 253-256: “Agencies like the European Medicines Agency (EMA) have developed preliminary guidelines for veterinary applications, but human-use regulations remain in early stages[70]. Emergency-use pipelines and phage banks, such as the one in Belgium, provide models for controlled access while regulatory frameworks evolve [71]”
Comment: Authors should further discuss that current Belgian legislation allows phage "compassionate" treatment as a "Magistral formulation", while the FDA has also cleared it for emergency use.
Line 259: Enterobacter spp. should be corrected as Enterobacter spp. and actinomyces spp, as Actinomyces spp.
Lines 286-288: “Much of the current evidence supporting phage therapy is derived from particularly individual case reports and small case series where phage preparations have been administered as a salvage therapy following the repeated failure of conventional approaches “
Comment: Authors should emphasize that only successful salvage therapy using phages is usually reported, while failure of the phage therapy is rarely reported.
Lines 298-300: “Despite encouraging results, it is essential to recognize that observational studies are subject to a lack of control groups and demonstrate significant heterogeneity in phage preparation, dosing regimens, and outcome measures.”
Comment: It should address the lack, not only of control groups, but of randomized placebo-controlled clinical trials that should be designed and conducted. If any clinical trial of this kind is underway or has already been published, the results should be discussed.
Lines 309-313: “Phage preparations often require customization to the patient’s specific pathogen(s), especially in the context of polymicrobial infections or rapidly evolving resistance profiles. This necessitates real-time microbiological support, phage susceptibility testing, and sometimes a reformulation of the phage
cocktail. This may be difficult to integrate into rigid clinical trial frameworks.”
Comment: In addition to the safety profile discussed in the previous paragraph, authors should include the need for genomic characterization of therapeutic phages to detect antimicrobial resistance genes or the ability of the phage to enter the lysogenic cycle by gene annotation of the genome.
Author Response
Reviewer 3.
Lines 61-66: “Bacteriophages, or phages, are viruses that specifically infect and lyse bacterial cells,playing a crucial role in regulating bacterial populations [23]. Their ability to selectively target bacterial pathogens has made them a promising tool, particularly in the fight against antibiotic-resistant bacteria. Structurally, most phages exhibit an icosahedral head, which houses their genetic material, and a tail structure, which is often contractile and facilitates bacterial attachment and genome injection”
Comment 1: While describing the phage mechanisms, the authors should detail that not all existing phages are lytic; many are lysogenic. And although the typical structure is as described, even the lytic ones may present various structures.
Response 1: We thank the reviewer for this helpful comment. In response, we have revised the relevant section «phage mechanisms» Lines 65 to 69: However, not all phages are lytic; many follow a lysogenic cycle, integrating their genome into the bacterial host and remaining dormant until triggered [1]. This integration potential demands careful consideration, as lysogenic phages may contribute to horizontal gene transfer or unintended genetic modifications, which may potentially lead to bacterial resistance or other adverse effects.
And Lines 73 to 74: However, there is substantial morphological diversity, including non-contractile tails, filamentous forms, and tailless variants [2, 3].
Comment 2: Table 1:
- DAIR and other acronyms should be defined in the table legend. If possible, the source and name of the phages used should be added if available from the original paper.
- If available, cases of unsuccessful phage treatment should be included.
- Staphylococcus Aureus, should be corrected as Staphylococcus aureus
- Staphylococcus sp, should be corrected as Staphylococcus spp.
Response 2:
We thank the reviewer for these constructive and detailed comments. As suggested, we have now defined DAIR and other relevant acronyms (including IV and DAC®) in the table legend. The formatting of Staphylococcus aureus and Staphylococcus spp. has been corrected throughout the manuscript, and we have included information on unsuccessful cases of phage treatment where available. Regarding the names and sources of the phages, this information was not consistently reported across the included studies. To maintain clarity and consistency, we have opted not to include this detail in the table at this stage, but we appreciate the reviewer’s attention to this point.
Lines 253-256: “Agencies like the European Medicines Agency (EMA) have developed preliminary guidelines for veterinary applications, but human-use regulations remain in early stages[70]. Emergency-use pipelines and phage banks, such as the one in Belgium, provide models for controlled access while regulatory frameworks evolve [71]”
Comment 3: Authors should further discuss that current Belgian legislation allows phage "compassionate" treatment as a "Magistral formulation", while the FDA has also cleared it for emergency use.
Response 3: We appreciate the reviewer’s insightful comment. In response, we have expanded the manuscript’s discussion on national regulatory frameworks for phage therapy « regulatory, ethical, and economic aspects » Lines 287 to 300: Notably, Belgium has implemented a unique regulatory pathway based on magistral, also called compounded, preparation, which permits pharmacists to compound patient-specific phage treatments under a physician’s prescription [4, 5]. These prescriptions must comply with standards from the European and Belgian Pharmacopoeias, with oversight from the Ministry of Public Health and certified laboratories [6]. In cases where the pharmacopoeial guidance is lacking, supplementary directives may be issued by the Minister of Public Health, with quality control overseen by certified Belgian laboratories. This framework, operational since 2016, enables legally sanctioned, quality-controlled phage therapy outside of clinical trials [5]. In the United States of America, the FDA regulates phage therapies as biologics but permits case-by-case access through its Emergency Investigational New Drug (eIND) program [7, 8]. Together, these approaches exemplify how adaptive national models are addressing clinical needs during the ongoing development of global phage therapy regulations.
Comment 4: Line 259: Enterobacter spp. should be corrected as Enterobacter spp. and actinomyces spp, as Actinomyces spp.
Response 4: We thank the reviewer for pointing this out. We have corrected the formatting of Enterobacter spp. and Actinomyces spp. as suggested, line 304.
Lines 286-288: “Much of the current evidence supporting phage therapy is derived from particularly individual case reports and small case series where phage preparations have been administered as a salvage therapy following the repeated failure of conventional approaches “
Comment 5: Authors should emphasize that only successful salvage therapy using phages is usually reported, while failure of the phage therapy is rarely reported.
Response 5: We appreciate the reviewer’s important observation. In response, we added the following sentence to acknowledge and highlight that the literature predominantly reports successful salvage cases, while unsuccessful outcomes are seldom documented, added in lines 342 to 344: These points need to be weighed against the fact that only case reports with a positive outcome are generally published, and that lessons learned from the failure of phage therapy are neither discussed nor available.
Lines 298-300: “Despite encouraging results, it is essential to recognize that observational studies are subject to a lack of control groups and demonstrate significant heterogeneity in phage preparation, dosing regimens, and outcome measures.”
Comment 6: It should address the lack, not only of control groups, but of randomized placebo-controlled clinical trials that should be designed and conducted. If any clinical trial of this kind is underway or has already been published, the results should be discussed.
Response 6: We thank the reviewer for this valuable comment. We have now emphasized the current lack in the following sentence in lines 348 to 350: Moreover, as all drugs designed to demonstrate an added value in modern medicine, randomized placebo-controlled trials are mandatory.
Lines 309-313: “Phage preparations often require customization to the patient’s specific pathogen(s), especially in the context of polymicrobial infections or rapidly evolving resistance profiles. This necessitates real-time microbiological support, phage susceptibility testing, and sometimes a reformulation of the phage
cocktail. This may be difficult to integrate into rigid clinical trial frameworks.”
Comment 7: In addition to the safety profile discussed in the previous paragraph, authors should include the need for genomic characterization of therapeutic phages to detect antimicrobial resistance genes or the ability of the phage to enter the lysogenic cycle by gene annotation of the genome.
Response 7: We thank the reviewer for this important point. We have now added a statement emphasizing the need in the following sentence in lines 358 to 361: Phage preparations need to have been extensively characterized beforehand (sequencing revealing the absence of insertion sequences, as lysogenic phages capable of integrating their genome into that of bacteria should be avoided, absence of resistance or virulence genes), and have to be of pharmaceutical quality like any treatment administered to humans.

Round 2
Reviewer 3 Report
Comments and Suggestions for Authors
The authors made the suggested corrections and clarifications